# The Impact of ETV6-NTRK3 Oncogenic Gene Fusions on Molecular and Signaling Pathway Alterations

**DOI:** 10.3390/cancers15174246

**Published:** 2023-08-24

**Authors:** Matias Kinnunen, Xiaonan Liu, Elina Niemelä, Tiina Öhman, Lisa Gawriyski, Kari Salokas, Salla Keskitalo, Markku Varjosalo

**Affiliations:** 1Institute of Biotechnology, University of Helsinki, 00014 Helsinki, Finland; 2Helsinki Institute of Life Science, University of Helsinki, 00014 Helsinki, Finland

**Keywords:** ETV6-NTRK3, gene fusion, breakpoint variant, proteomics, interaction analysis, mass spectrometry, biotin proximity labeling, BioID

## Abstract

**Simple Summary:**

Gene fusions produce chimeric fusion proteins with unpredictable properties, many of which function as oncogenic drivers. ETV6-NTRK3 produces a constitutively active fusion kinase, which is not membrane bound like the endogenous NTRK3 kinase. Several studies have investigated the interactions of ETV6-NTRK3 over the years with few interactors being reported and some interactors reportedly being lost compared to the endogenous NTRK3. We utilize proximity labeling to produce the first proteomics level study on ETV6-NTRK3 close protein milieu and analyze the interactomes of all the four known ETV6-NTRK3 variants with functional kinase domains. Existing clinical literature shows that these four ETV6-NTRK3 variants occur at differing frequencies and originate in different tissues, with a “canonical” variant being more frequent and is also reported in other tissues. The ETV6-NTRK3 variants were found to have both similarities and differences in interactors, which might explain differences in reported frequencies and tissue specificities of the variants.

**Abstract:**

Chromosomal translocations creating fusion genes are common cancer drivers. The oncogenic ETV6-NTRK3 (EN) gene fusion joins the sterile alpha domain of the ETV6 transcription factor with the tyrosine kinase domain of the neurotrophin-3 receptor NTRK3. Four EN variants with alternating break points have since been detected in a wide range of human cancers. To provide molecular level insight into EN oncogenesis, we employed a proximity labeling mass spectrometry approach to define the molecular context of the fusions. We identify in total 237 high-confidence interactors, which link EN fusions to several key signaling pathways, including ERBB, insulin and JAK/STAT. We then assessed the effects of EN variants on these pathways, and showed that the pan NTRK inhibitor Selitrectinib (LOXO-195) inhibits the oncogenic activity of EN2, the most common variant. This systems-level analysis defines the molecular framework in which EN oncofusions operate to promote cancer and provides some mechanisms for therapeutics.

## 1. Introduction

Chromosomal translocations creating fusion genes are among the most common mutation class of known cancer genes [1]. Recently, oncogenic fusion genes (oncofusions, OFs) have been found in many hematological and solid tumors, demonstrating that translocations are a common cause of malignancy [2,3]. The frequency of recurrent gene fusions varies depending on the specific type of cancer, but currently identified translocations are estimated to drive up to 20% of cancer morbidity [4,5,6,7].

The COSMIC (catalogue of somatic mutations in cancer) database [8] lists 300 unique gene fusion pairs found in cancers. Of these fusion pairs, 213 have been reported only in a single type of cancer and few are reported in several cancer types. According to the cosmic database, ETV6-NTRK3 (EN) is reported in seven different cancers, and while there are six fusion pairs that are reported in larger number of different cancers, EN harboring malignancies are also reported in seven different tissue sites, which is more than any other listed fusion pair. EN also belongs to a small group of fusion pairs that are reported in both solid and hematological malignancies.

ETV6 forms fusions with many kinases, except for EN, which are reported in either hematological or solid malignancies, and EN further distinguishes itself by being reported in several subtypes in both cases [9]. The COSMIC database currently lists NTRK3 fusion only with ETV6, but there is mounting evidence for more NTRK3 fusions with the following partners: SQSTM1, EML4, MYO5A, SPECC1L, TFG, RBPMS and STRN [10,11,12,13,14,15,16,17,18,19,20,21]. EN is known to activate ERK and AKT signaling, and AP-1 (JUN/FOS) activation is implicated in breast cancer initiation; also, the EN relies on functional IGF1R signaling in the host cell [22,23,24].

ETV6-NTRK3 (EN) oncofusion, is a product of the chromosomal t (12; 15) (p13; q25) translocation (Figure 1A), which fuses the N-terminal SAM (sterile alpha motif) domain of ETV6 to the C-terminal protein tyrosine kinase domain of NTRK3 (also known as TrkC). EN is expressed from the ETV6 promoter in the fused chromosome 15 (Figure 1A). The ETV6 promoter is generally more active, than NTRK3 promoter, and causes EN fusions to be more highly expressed in several tissues and especially in bone marrow and salivary glands, compared to NTRK3 (Figure 1B).

While mentioning EN, it is important to bear in mind that it is not a single fusion but has several variants where the N–terminal of ETV6 and C-terminal of NTRK3 fuse at different break points, combining different exons. Four different EN fusions have been characterized, which we refer to as EN1–4 from the longest to the shortest variant (Figure 1C). The fused fragments are either ETV6 up to exon 4 or 5 fused to NTRK3 exons from 12 or 13 onwards. EN2 (5–13) is the most common and is considered the “canonical” variant. The variant EN4 (4–15), in contrast, has been reported only in leukemia and is therefore called the “leukemia variant” [25]. EN1 (5–13) and EN3 (4–13) were both first reported in papillary thyroid carcinoma [26], where EN3 is more common than EN1. The fusion sites alternate between ETV6 exon 5, which forms the interdomain between the ETV6 SAM and transactivation domains, and NTRK3 exon 12, which forms part of the sequence between the transmembrane domain and the beginning of the kinase domain. ETV6 exon 5 is reported to interact with proteins involved in transcriptional repression: NCoR, mSin3 and SMRT [27,28], whereas the NTRK3 exon 12 site Y516 mediates interactions with SHC1, GRB2 and PI3K p85 [29]. Presence of these domains therefore influences the effects of EN fusions on downstream signaling, and may explain the cancer-specific effects of EN variants. The EN2 and EN4 variants, for example, lack exon 12 of NTRK3, and the loss of SHC1, GRB2 and PI3K p85 interaction has been confirmed using immunoprecipitation for EN2 [30]. However, although EN2 lacks the SHC1 binding site, it nevertheless still activates the Ras/ERK and PI3K/AKT pathways [23], likely involving a stable interaction with IRS adaptors [31]. Though, IRS adaptors can recruit factors like GRB2, PI3K p85 and SHIP2 [32,33,34], which so far have not been found to interact with EN [23,30]. Thus, our understanding of variant-specific interactions and downstream signaling effect is still incomplete.

EN fusions also display the alternative splicing of NTRK3 exon 16 [35], also known as the NTRK3 kinase insert (i) (Figure 1C, highlighted in red). The insert is located next to a major auto-phosphorylation motif that is unique to NTRK3 in the NTRK family [35]. Like the exons described above, inclusion of the insert impacts downstream signaling, in this case inhibiting the PI3K, MAPK and PLCγ pathways and inducing FOS and MYC expression [29,35]. The absence of the kinase insert therefore enables NTRK3 to transform cells and induce proliferation in response to the NT-3 ligand [29,35,36]. A previous study compared the insert-containing EN2 variant (EN2+i) to the EN4 variant, and it found both variants were auto-phosphorylated and both enabled growth of Ba/F3 cells in growth factor-depleted media, but EN4 induced more growth than EN2+i [37]. The study also showed EN4 to transform NIH3T3 but not EN2+i. The insertless EN mRNAs have also been shown to be more abundant in congenital mesoblastic nephroma and congenital fibrosarcoma [38,39]. However, not much is known about the role of the insert regarding the interactome.

To better understand the effects of EN variants and the impact of the insert, we now comprehensively compare the interactomes of EN oncofusion variants to those of the wild-type ETV6 and NTRK3 proteins. Using proximity labeling coupled with mass spectrometry, we identify 153 fusion specific interactions, linked to the ERK, STAT, JNK and AKT pathways. Using pathway-specific, luciferase-based reporter assays and analysis of phosphorylated proteins, we confirm the effect of the different EN variants on the activation of these signaling pathways. In addition, we show that the pan NTRK inhibitor Selitrectinib (LOXO-195) suppresses fusion-specific interactions for the most common oncogenic EN2 variant. Selitrectinib restores ERK, STAT1, STAT3 and JNK pathway activation to normal levels, and prevents EN2- and EN4-induced cellular transformation. This first systems-level analysis of the ETV6-NTRK3 oncogenic gene fusions provides a wealth of information on molecular interactions and pathways induced by these proteins, and offers insights into the cancerogenic effects of these fusions.

## 2. Materials and Methods

### 2.1. Cell Culture

Flp-In™ T-REx™ 293 (Thermo Fisher Scientific #10270-106, Waltham, MA, USA) cells were cultured in low-glucose DMEM pH 7.4 supplemented with penicillin–streptomycin and 10% FBS (Gibco #10270-106, Billings, MT, USA) and maintained at 37 °C and 5% CO_2_.

### 2.2. Cell Line Generation

Cell lines were generated from Flp-In™ T-REx™ 293 (Thermo Fisher Scientific #10270-106) by transfecting the cell lines with the gene of interest (GOI) in MAC-tag plasmids (Addgene, #108078 and #108077, Watertown, MA, USA) and pOG44 Flp-Recombinase Expression Vector (Thermo Fisher Scientific #V600520) in a 1:5 ratio. Positive cells were selected in 0.1 mg/mL hygromycin B (Thermo Fisher Scientific #10687010).

### 2.3. Biotin Proximity Assay

Generated stable cells were induced, at 70% confluency, with 2 μg/mL tetracycline and 50 μM biotin 24 h prior to harvesting. Cells were washed with ice cold 100 mM CaCl_2_ and 100 mM MgCl_2_-PBS and detached with ice cold 1 mM EDTA-PBS. A total of 1.5 × 10^8^ cells were harvested per sample, and the samples were prepared in triplicates.

Pellets were lysed, and lysates were sonicated in 3 mL of ice-cold lysis buffer (1% n-Dodecyl-β-D-Maltoside, 50 mM HEPS, 5 mM EDTA, 150 mM NaCl, 50 mM NaF pH 8.0 supplemented with 1 mM DTT, 1 mM PMSF, 1.5 mM NaVO_4_, 1× Sigma protease inhibitor cocktail, 0.1% SDS and 250 U benzonase). Insoluble cell debris was removed using centrifugation, and the biotinylated proteins were purified with strep-tactin sepharose beads (IBA Lifesciences #2-1201-025, Göttingen, Germany) in spin columns (Bio-Rad #7326008, Hercules, CA, USA). A 200 μL volume of strep-tactin beads were prewashed with 1ml of lysis buffer. After loading the samples, the beads were washed with 3× 1 mL of lysis buffer, followed by 4× 1 mL of 50 mM HEPS, 5 mM EDTA, 150 mM NaCl and 50 mM NaF pH 8.0. Samples were eluted by incubating the beads twice for 5 min in 300 μL of 50 mM HEPS, 5 mM EDTA, 150 mM NaCl and 50 mM NaF pH 8.0 with 0.5 mM biotin.

Cysteine bonds were reduced with 5 mM Tris (2-carboxyethyl)phosphine (TCEP) for 30 min at 37 °C and subsequently alkylated with 10 mM iodoacetamide for 30 min at room temperature. Proteins were digested to tryptic peptides with modified trypsin (Promega, #V5113, Madison, WI, USA) at 37 °C overnight. Digestions were quenched with 10% trifluoroacetic acid (TFA), and samples were desalted using C18 reversed-phase spin columns according to the manufacturer’s instructions (Harvard Apparatus #74-4601, Holliston, MA, USA). Purified peptides were dried in a vacuum centrifuge and reconstituted into 30 μL of 0.1% TFA and 1% acetonitrile.

### 2.4. Mass Spectrometry Analysis

The analysis was performed on a Q Exactive™ Hybrid Quadrupole-Orbitrap Mass Spectrometer (Thermo Fisher Scientific) using Xcalibur version 2.0.7 SP1 (Thermo Fisher Scientific) coupled with an EASY-nLC 1000-system via an electrospray ionization sprayer (Thermo Fisher Scientific). For each sample, three biological replicates were used, and a 4 μL peptide sample was loaded for each analysis. Peptides were eluted and separated with a C-18-packed pre-column and an analytical column, using a 60 min buffer gradient from 5 to 35% buffer B (buffer B: 0.1% formic acid in 98% acetonitrile and 2% HPLC-grade water), followed by a 5 min gradient from 35 to 80% buffer B, and a 10 min gradient from 80 to 100% buffer B at a flow rate of 300 nL/min (buffer A: 0.1% formic acid in 2% acetonitrile and 98% HPLC-grade water). Peptides analysis was performed in a data-dependent acquisition mode using FTMS full scan (200–2000 *m/z*) resolution of 70,000 and higher-energy collision dissociation (HCD) scan of the top 20 most abundant ions.

For protein identification, mass spectrometry RAW files were searched against a library of the Uniprot human protein sequences and high-confidence identifications (FDR 0.01) were performed using Percolator (Thermo Fisher Scientific). High-confidence PPIs were filtered using SAINT [40] and GFP BioID control samples. FDR threshold of 0.03 was used to filter BioID. CRAPome contaminant repository [41] was used to filter out interactors that were observed in over 20% of experiments (82/411). Proteins identified in over 20% CRAPome experiments were still retained in the results if the resulting PSM average value was both at least three times higher than the CRAPome average and at least as high as the CRAPome maximum PSM value.

### 2.5. Heatmap

The SAINT filtered PPI (protein–protein interaction) data was used to generate a heatmap. Individual triplicate sample PSM values were normalized using MAC-tag abundance (normalized for Strep-HA peptide spectral matches of MAC-tag values) prior to calculating the PPI average PSM values. The heatmap was generated with Heatmapper [42] with LOG2 values, using Pearson average linkage.

### 2.6. Dotplot and Correlation Plot

Dotplots and correlation plots were generated using ProHits-viz website [43] dotplot and correlation analysis tools.

### 2.7. GO Term Analysis

High-confidence interactors obtained from our study were subjected to gene ontology (GO) analysis (DAVID analysis [44], KEGG pathway database [45] and Reactome pathway-based enrichment analysis [46]). GO term fusion was used, and only most relevant and enriched terms with *p* values ≤ 0.05 were displayed.

### 2.8. Luciferase Assays

2 × 10^5^ HEK293 cells were seeded on clear flat-bottom 96-well plates (Corning #3610, Corning, NY, USA). Each well was transfected on the next day with 50 ng of luciferase reporter constructs from the Cignal reporter assay kit (Qiagen #336841, Hilden Germany), 2.5 ng of plasmid expressing Renilla luciferase transfection control and 47.5 ng of plasmid expressing protein of interest. Incubation time after transfection was 16 h for STAT3 and 30 h for both AP-1 and ERK pathways. After the incubation, the media was removed, and cells were lysed in 25 μL 1× passive lysis buffer (dual luciferase kit) for 25 min in a shaker at RT. Luciferase readings were recorded on a CLARIOstar plate reader (BMG Labtech, Ortenberg, Germany) after injecting 25 μL of the corresponding luciferase reagents. Both Firefly luciferase and Renilla luciferase readings were measured 11 times over a 7 s time span, and the average reading values were used to calculate the results. The samples were performed in quadruplicates, and the transfection rates between replicates were normalized by dividing the luciferase reading with the Renilla luciferase reading.

### 2.9. Barcoded Mass cytometry

Stable cell lines generated from Flp-In™ T-REx™ 293 (Thermo Fisher Scientific) were seeded at a 1 × 10^6^ concentration per 10 cm culture plate. Cell lines were induced at 50% confluency with 2 μg/mL tetracycline, except for the control. Cells were trypsinized and counted by FACS using Guava easyCyte (Millipore, Burlington, MA, USA) after 24 h incubation. In a fresh tube, 3 × 10^6^ cells were added and stained with Cisplatin as a viability stain. Each individual cell line tube was stained with a Cell-ID 20-Plex Pd Barcoding Kit antibody (Fluidigm #201060, San Francisco, CA, USA) after which 5 × 10^5^ cells from six individual samples were pooled into one tube with 3 × 10^6^ cells total. The barcoded pooled samples were then stained with an antibody cocktail from the Maxpar^®^ Signaling I phospho-specific antibody panel (Fluidigm #201309). Stained barcoded cell suspension pools were run on the Helios mass cytometer (Fluidigm) until 300,000 events were recorded. Results were normalized prior to de-barcoding and results were analyzed in the Cytobank cloud service. Mean values of phosphorylation events were exported and phosphorylation events between different cell lines were normalized to the values of EN2 cell line without tetracycline induction.

### 2.10. Blots

Stable cell lines generated from Flp-In™ T-REx™ 293 (Thermo Fisher Scientific) were seeded at a 1 × 10^6^ concentration per 10 cm culture plate. Cell lines were induced at 50% confluency with one of three treatments: 2 μg/mL tetracycline and 100 mM inhibitor Selitrectinib/LOXO-195 (MedChemExpress #HY-101977, South Brunswick Township, NJ, USA), 2 μg/mL tetracycline and DMSO equivalent of inhibitor or low-glucose DMEM with DMSO equivalent of inhibitor (control sample). Cells were washed with ice-cold PBS and lysed in ice-cold Ripa buffer (150 mM NaCl, 0.5% deoxycholate, 1% Igepal, 0.1% SDS pH 7.4) supplemented with 1mM PMSF and 1x protease inhibitor cocktail (Sigma-Aldrich #P8340, St. Louis, MO, USA).

Cells were detached into the lysis buffer with a cell scraper, and lysate was pipetted into fresh 1.5 mL tubes. Lysates were incubated 30 min on ice, and the samples were centrifuged at 16,000× *g* for 15 min at +4 °C, to remove insoluble debris. The clear lysate was then mixed with a 5× laemmli sample buffer and incubated at 95 °C for 5 min. 5 μL of lysate SDS-PAGE samples were loaded on precast SDS-PAGE gels (Bio-Rad #4561034) and transferred onto nitrocellulose membrane (PerkinElmer #NBA085C001EA, Waltham, MA, USA) with semi-dry transfer (Bio-Rad #1703940). The membranes were blocked with 5% milk–in 0.05% Tween–TBS. HA antigen was detected with primary HA antibody (BioLegend #16B12, San Diego, CA, USA, 1:2000 dilution in blocking solution). Phosphorylated ERK1/2 was detected using a primary pERK1/2 (T202/Y204) antibody (Cell signaling technology #9101, 1:1000 dilution in 5% BSA—in 0.05% Tween). The primary antibodies were detected with a secondary antibody coupled with HRP (GE HealthCare #NA931, Chicago, IL, USA, 1:1000 dilution in blocking solution), and ECL reaction (Amersham #RPN2232, Buckinghamshire, UK) was developed on photographic films (FujiFilm #47410, Minato city, Tokyo, Japan) or imaged using an iBright 1500 imaging system (Thermo Fisher Scientific).

### 2.11. Morphology Microscopy

At 60% confluency, stable cell lines were induced to express the insertless ETV6-NTRK3 fusions by adding 2 μg/mL tetracycline. The cells were imaged 24 h after induction, using a 10× objective lens.

### 2.12. Fluorescence Microscopy

HEK293 cells were grown on coverslips and transfected with corresponding plasmids. After 24 h, cells were fixed with 4% paraformaldehyde for 20 min and permeabilized with 0.1% Triton X-100 in PBS. For antibody staining, permeabilized cells were blocked with Dulbecco plus 0.2% BSA for one hour, incubated with primary antibody for one hour, and incubated with fluorescence-conjugated secondary antibody for one hour at room temperature. Cells were mounted in moviol supplemented with DABCO, and all imaging was performed using a TCS SP8 STED confocal microscope (Leica, Wetzlar, Germany), using the 93× glycerol immersion objective lens.

## 3. Results

### 3.1. EN Variants Are Present in Different Human Cancers and Vary in Their Ability to Transform Cells

To gain further insight into the unique molecular mechanisms underlying the effects of different EN variants, and the contexts in which they act, we assessed the relative incidence of EN variants across multiple human cancers. We performed a thorough literature search of papers describing the variants in different human cancers (Appendix A). Variant frequency was calculated from the reported cases in which the corresponding EN variant could be definitively identified (Figure 1D and Appendix A). EN2 was most frequently identified in analogous secretory tissue carcinomas such as those in breast [47] and salivary glands [48], followed by congenital fibrosarcoma (CFS) and congenital mesoblastic nephroma (CMN), but was also reported in inflammatory myofibroblastic tumor (IMT), acute myeloid leukemia (AML) [49], acute lymphoblastic leukemia (ALL) [50,51], chronic eosinophilic leukemia (CEL) [52], sinonasal adenocarcinoma (SNAC) [53] and glioma [54]. EN1 and EN3 were first reported in papillary thyroid carcinoma (PTC) [26] and in gastrointestinal stromal tumor (GIST) [55,56]. EN3 has additionally been reported in glioma [57] and secretory carcinomas [58,59], while EN1 has also been reported in AML [60]. EN4, the EN “leukemia variant”, has been reported in AML [25] and cell lines of acute promyelocytic leukemia (APL) and acute myeloid leukemia (AML) [61,62]. There are also mentions of kinase domain truncated variants, along with their untruncated transcripts, EN2 in CFS and AML [38,49] as well as EN4 in AML [25].

To assess whether general characteristics of these variants, such as expression level and transformation ability, correlated with their differing cancer presentations, we examined the effects of EN variants in HEK293 Flp-In cells. We constructed cell lines stably expressing inducible MAC-tagged [63] NTRK3, NTRK3+i, ETV6, EN and EN+i variants. All cell lines were isogenic, and induction protein expression levels were controlled to prevent overexpression. We found that the EN+i variants (containing the insert) maintained lower protein amounts compared to the EN variants, with the exception of EN1+i (Figure 1E). However, we did not observe a notable difference in abundance between the wild-type NTRK3 with and without the kinase insert, perhaps due to the lack of NT-3 ligand. To assess the transformation potential of the EN variants, we examined the effect of the variants on the HEK293 Flp-In cell line morphology (Figure 1F). Without induction (i.e., without transgene expression), all cell lines display highly similar cell morphology and densities. However, upon induction by tetracycline, cells expressing the EN2 and EN4 variants dramatically change their morphology, while cells expressing EN1 and EN3 do not, suggesting that expression of EN2 and EN4 is sufficient to drive cell transformation. This initial examination of the EN variants showed that EN variants are disparately reported in clinical cancer cases and the kinase insert has an effect on the EN variants protein levels and that the EN variants appear to differ in their capacity to transform the generated stable cell lines.

### 3.2. Interactome Analysis Reveals Newly Acquired Molecular Interactions of EN Oncofusions and Identifies Signaling Pathway Targeted by EN Variants

EN fusion proteins have been shown to activate specific signaling pathways, but a complete understanding of their impact on molecular pathways is still lacking. To gain further insight into the mechanisms by which EN oncofusions promote cellular transformation, we employed an unbiased biochemical method to define the global interactomes of EN variants in comparison to that of the wild-type proteins. We used the MAC-tag system and employed proximity labeling mass spectrometry in the generated stable HEK293 Flp-In cell lines (Figure 2A) [63,64]. In this approach, the tagged transgene expression is induced by tetracyclin, and the biotin ligase BirA tag is activated by adding biotin to the cell culture. During the following 24 h, the transgenes are expressed, and they biotinylate nearby protein lysine residues. Biotinylated proteins can subsequently be purified from lysate to identify proteins which were near the transgene products while the cells were still intact. After stringent statistical filtering, we identified over 2000 specific interactions: EN1+i (173), EN2+i (188), EN3+i (204), EN4+i (148), EN1 (125), EN2 (165), EN3 (168), EN4 (140), NTRK3+i (394), NTRK3 (306) and ETV6 (105) (Appendix A). We performed a correlation analysis of all baits used in this study (Figure 2B). The two NTRK3 baits clustered together and ETV6 was distinct from all the other baits. The correlation between the EN and EN+i clusters was also strong. Moreover, the interactome profiles of the four EN and four EN+i variants were highly correlated within their own groups. Thus, EN fusions show consistent molecular interactions, which, as a whole, are distinctly separate from both the wild-type NTRK3 and ETV6.

To understand how EN fusions alter the binding landscape of NTRK3 and ETV6, we studied the subcellular localization of the proteins and their interactomes. We used our MS interactors-based methodology, which utilizes quantitative interactome information to infer the cellular distribution of the tested bait proteins [63] (Figure 2C). In general, the wild-type NTRK3 protein and the NTRK3+i variant exhibited a similar distribution, showing strong association with plasma membrane proteins, and a somewhat lower association with cellular organelles including the Golgi, lysosome and ER. ETV6 showed a different distribution, highly focused on the Golgi, chromatin and nuclear envelope, with weaker association with intermediate filaments, focal adhesion, plasma membrane and nucleoplasm. The distribution of the EN variants was distinct from the wild-type proteins, though with some overlap. EN+i variants showed the strongest association with the plasma membrane, followed by the cytoskeleton (“actin filament”) and nucleolus, cell junction, intermediate filaments and cell junction, and weaker associations with focal adhesion, exosome, proteasome, mitochondria and microtubules, with some variants also showing association with the endosome, ER, Golgi and the centrosome. The distribution of EN variants showed some similarity to that of the EN+i variants, but with somewhat different strengths. EN+i variants most strongly associated with the nucleolus, plasma membrane and cell junction, followed by focal adhesion, actin filament, intermediate filament and microtubules, with a few variants showing weak association with the exosome, ER, Golgi, mitochondrial and centrosome. One noteworthy difference is the “proteasome” localization, specific for the EN+i variants, and “lysosome” localization, specific to NTRK3. The component that linked the EN+i variants strongly to the proteasome was PSA4, with the EN4+i variant also interacting with another proteasome-associated protein, PSB1. Both of these proteins are part of the proteasome 20s core particle [65]. These association may reflect the targeting of these variants for degradation. Overall, these results suggest that EN variants may be targeted to different subcellular regions, and this differential localization may impact their interaction landscape, their effect on different signaling pathways and their vulnerability to degradation.

These interactions are consistent with some previous work on EN proteins, albeit with a few notable differences. For example, interactions with SHC1 and IRS2 and STAT1 (Figure 2D) have been tested in existing literature [31,66]. Consistent with this work, our data show that all of the EN variants interact with STAT1, and all NTRK3+i interact with IRS2. Interestingly, we also detected unreported interactions of STAT family members: STAT3 with EN1+i and STAT5B with EN1+i, EN2+i and EN4+i (Figure 2D). A previous study found no SHC1 interaction with EN2 likely due to not having the NTRK3 Y516 binding site [30], and our data suggest that all EN variants, including those missing the Y516 site, interact with SHC1. This difference may be due to the extremely sensitive proximity labeling approach, which can detect weak and transient interactions. Consistent with this hypothesis, the SHC1 interaction was stronger for variants harboring the NTRK3 Y516 site than those without. Both the detected SHC1 and IRS2 adaptors can recruit GRB2 and SOS, to mediate the ERK [34] signaling and GAB adaptors to mediate PI3K-AKT signaling [67,68]. Additionally, specifically IRS2 peptides have been shown to recruit SHC1 as well [32]. GRB2 and SOS were also detected as EN interactors but not GAB. The GRB2 interaction pattern is mostly similar to that of SHC1 and IRS2, but the SOS interaction appeared strongly dependent on the presence of the kinase insert. The SOS interaction was strongest for the EN1+i, EN2+i and EN3+i variants but barely detectable with their insertless counterparts. These interactions underline the likely connection between EN variant and the ERK and PI3K signaling pathways, and highlight the potential impact of the insert on EN interactions.

To examine further how the kinase insert affects the interactome and function of EN variants, we hierarchically clustered the eight EN variants based on their interactors (Figure 2E). To obtain a clearer picture, we restricted the heatmap to interactions involving all four variants from either group. The resulting heatmap can be divided into three types of interactor clusters: (1) primarily EN-interacting, (2) primarily EN+i-interacting and (3) interactors shared by both groups. Next, we applied Reactome pathway enrichment analysis on these clusters, filtering the pathway terms with *p*-value < 0.05 (Appendix A) and summarizing the 20 terms with the lowest *p*-values for each cluster (Appendix A). The analysis identified three major pathways for the “EN cluster”: cell cycle (8), signaling by kinases (5) and signaling by Rho GTPases (4). The “shared cluster” enriched terms were mostly related to cell cycle (6), Rho GTPase signaling (7) and kinase signaling (5). The “EN+i cluster” interactors were associated with kinase signaling (18) and Rho GTPase signaling (2). This comparison showed that group-wise the interactors highly enrich associations with kinase signaling and the cell cycle, and Rho GTPases and the EN variants have stronger associations with cell cycle compared to the EN+i variants. Surprisingly, the EN+i cluster interactors were not enriched with cell cycle-related terms.

### 3.3. Analysis of the EN Interactome Map Reveals Connections to Multiple Signaling Pathways

To better understand the physical and functional relationships among oncogenic EN variants, we constructed an interaction network map (Figure 3A) showing 234 protein–protein interactors (PPI) from the four EN variants (individually EN1 125 PPIs, EN2 165 PPIs, EN3 167 PPIs, EN4 139 PPIs). Of those, 153 are oncofusion-specific, meaning they were not observed with the wild-type ETV6 or NTRK3 (Appendix A). The four EN variants share 58 interactions with NTRK3 and 19 interactions with ETV6, with 4 interactions shared with both ETV6 and NTRK3. The interaction map underlines the large number of new interactions induced by formation of EN fusion proteins.

Consistent with the earlier interaction data, the interaction network map (Figure 3A) also highlights the association of EN fusions with multiple signaling pathways through interactions with 24 adaptor/scaffold proteins, 23 protein kinases, 10 protein phosphatases, 20 Guanine Exchange Factors and GTPase activating proteins and 5 transcription factors. Additionally, interactions with 34 cell adhesion/cytoskeleton, 6 cilia, 8 transport, 14 centrosome, 14 cytokinesis and 14 translation associated proteins were detected. The interactomes also contained five uncharacterized proteins, of which KIAA1671 had some of the highest PSM (peptide spectrum match) values out of all interactors. The significance of this interaction will need to be examined in future studies.

One other notable property of the map is that it often includes proteins known to function together in protein complexes, as defined in the CORUM database (Appendix A). Examples include the adaptors TAB1 and TAB3, which are essential for the activation of the TAK1 (M3K7) kinase in the TAK1–TAB1–TAB3 complex [69]; SH3K1 (CIN85), which forms part of the CIN85 complex, whose components were also found to be interactors [70]; and SH3K1 and CBL, which together form the CIN85–CBL complex, mediating receptor tyrosine kinase downregulation [71]. Proteins regulating the ubiquitin ligase CBL were also identified as interactors (SHKB1 [72], UBS3B [73] and SH2B2 [74]), as were components of the CCNB2–CDK1 mitotic complex [75] and the CCNB1–CDK1 apoptotic complex [76]. Similarly, the interactome contained components of the c-Abl–CAS–Abi1 complex, which modulates actin [77], the GIT1–ARHGEF7–PAK1–PXN complex, which activates RAC1 [78], the AURKA–CKAP5–TACC1 chromatin complex [79] and the AXIN1–APC–CTNNB1–GSK3B Wnt-regulatory complex [80]. The presence of such functionally related proteins provides confirmation for the reliability of our data, and suggests a functional link between EN variants and the pathways and/or processes regulated by these complexes.

The interactome data also allowed us to interrogate the domains and motifs underlying the observed interactions. We used InterPro database to identify functional domains in interactors of the four EN variants (Figure 3B). Not surprisingly, the interactors contained many protein kinase-associated domains, kinase-interacting signaling SH2/SH3 domains and protein phosphatase domains. The interactors also contained pleckstrin homology-like (PH) domains, required for recruiting proteins into different membranes by phosphatidylinositols and targeting them to appropriate cellular compartments and/or signal transduction pathways [81]. We also detected enrichment of PDZ/LIM domains, known to play mechanosensory roles and to be associated with nuclear shuttling and transcription regulation [82]. The identity of these domains is consistent with the functional annotations derived earlier, and further supports the idea that EN fusion proteins impact cellular behavior via interactions with multiple kinase and signaling transduction pathways.

Indeed, Reactome (Figure 3C) and KEGG (Appendix A) pathway analyses of interactors identified multiple terms consistent with involvement in various signaling pathways, including p53 in cancer, cell cycle (G2/M DNA replication checkpoint, activation of NIMA kinases and phosphorylation of Emi1) and ERBB2 signaling. The ErbB signaling pathway was particularly enriched for interactors of the most common EN2 variant (*p* = 2.9 × 10^−9^). Other highly enriched signaling pathways associated with EN2 interactors included Neurotrophin (*p* = 3.5 × 10^−9^) and insulin signaling (*p* = 1.4 × 10^−8^). Interestingly, chronic myeloid leukemia was enriched in all but the EN3 variant, which is the only variant not reported in leukemias (Figure 1D). Thus, these data are consistent with known properties of EN variants, and underscore the association between EN variants and various signaling pathways, with potential relevance to cancer.

### 3.4. EN Variants Aberrantly Activate Several Key Signaling Pathways

To more directly examine the impact of EN variants on specific signaling pathways, namely, MAPK/ERK (ELK1), STAT (1,3,5), PI3K-AKT and JNK (AP-1), we employed pathway-specific luciferase reporter assays to examine pathway activation in response to EN expression (Figure 4A,C,D,E). We transiently expressed the EN variants, wild-type ETV6 and NTRK3 and GFP control in HEK293 cells. EN variants 1, 2 and 3 activated MAPK/ERK signaling with over 15-fold increase compared to control, whereas NRTK3 induced only small increase in activity, and EN4 exhibited an even smaller effect (Figure 4A). We validated these findings by measuring ERK (1/2) phosphorylation in the stable cell line lysates 24 h after induction (Figure 4B). The four EN variants, but not NTRK3, also strongly activated STAT3 (Figure 4D). STAT1, on the other hand, was mainly activated by EN1-3 and only modestly by NTRK3 (Figure 4C), mimicking the MAPK/ERK pattern. Interestingly, AP-1 was activated by all EN variants as well as NTRK3 and ETV6 (Figure 4E).

We also examined the phosphorylation events, in the stable cell lines, using mass cytometry and a phospho-specific antibody panel (Figure 4F). The antibodies on this panel target phosphorylated ribosomal protein S6 (pS325 and pS236) along with STAT1 (pY701), STAT3 (pY705) and STAT5 (pY694). For instance, the ribosomal protein S6 is activated, by phosphorylation, downstream of the PI3K-AKT pathway. Variants EN1-3 all increased phospho-S6 levels, with EN2 inducing pS6 five-fold more compared to ETV6 or NTRK3. All variants also induced pSTAT3 and pSTAT5, although the EN2 variant stood out with very strong induction of pSTAT5 and high pSTAT3 levels. Induction of pSTAT1 was relatively weak across all variants. These data confirm a positive effect of EN fusion proteins on PI3K and STAT pathway activation, although the level of induction varies across variants.

### 3.5. Selitrectinib Is a Potent Inhibitor of EN-Induced Transformation and Aberrant Pathway Activation

Selitrectinib (LOXO-195) is a recently developed, second generation small molecular inhibitor of tropomyosin kinase receptors. It also has low nanomolar inhibitory activity against NTRK1 and NTRK3, by inhibiting the autophosphorylation of the NTRK family kinases, and can inhibit NTRKs containing known drug resistance mutations [83]. We assessed whether this inhibitor could counteract the effects of EN variants on signaling pathway activation and cellular transformation.

We first tested the efficacy of Selitrectinib at preventing cellular transformation induced by expression of EN variants (Figure 5A). As shown above (Figure 1F), only variants EN2 and EN4 induced a significant change in the stable cell line morphology. When Selitrectinib (100 nM) was administered at the time of tetracycline induction, it prevented transformation and the cells retained their normal cellular morphology.

Focusing on EN2, we then tested how inhibition treatment affects the interactions. For this purpose, we performed interactome analyses in the presence and absence of Selitrectinib (Figure 5B and Appendix A). The inhibitor led to a significant (four-fold) decrease in the number of high-confidence interactors. We analyzed the enriched KEGG pathway associations of EN2 and EN2 treated with Selitrectinib (Appendix A), while EN2 interactome is associated with many cancer-related pathways, almost all these associations are lost after treatment with Selitrectinib. Of the 22 interactors detected in both the inhibited and uninhibited conditions, 16 were shared with either wild-type ETV6 or NTRK3. Therefore, Selitrectinib treatment specifically inhibits new interactions acquired by the EN2 variant. Next, we tested the effect of the inhibitor on protein expression levels. The Selitrectinib treatment clearly lowered EN1 and EN2 abundance in the stable cell lines, and completely diminished EN3 and EN4 to undetectable levels (Figure 5C). The inhibitor had no effect on the level of ETV6 but the abundance of NTRK3 was slightly lowered.

We studied the subcellular localizations of the constructs using confocal microscopy, with and without the inhibitor (Figure 5D). ETV6 was found mostly in the nucleus and was not noticeably different upon the inhibitor treatment. NTRK3 was found primarily in the cell periphery and around the nucleus, likely internalized on membranes. The inhibitor-treated NTRK3 appeared to be more on the cell periphery than in the uninhibited sample, possibly meaning that the uninhibited NTRK3 is slightly activated and internalized as a consequence. The EN variants were distributed largely in the cytoplasm. Inhibitor treatment caused the EN variants to localize into small speckles in the cytoplasm, showing possible transportation to proteasomes for degradation. The EN1 variant was least impacted by Selitrectinib but still showed a slightly more speckled cellular distribution in the presence of inhibitor. The inhibitor therefore leads to an altered subcellular distribution for all tested NTRK3 kinase domain containing proteins, with varying severity. EN1 localization under inhibition was seemingly the least effected, and the EN1+i variant also maintained higher protein abundances as shown above (Figure 1E). Consistent with the NTRK kinase inhibiting function of Selitrectinib, we found that the inhibitor prevented increased activity of the MAPK/ERK, STAT1 and STAT3 pathway in the presence of EN1-3, as measured by the reporter assays (Figure 5E). The effect on the AP1 was modest, although still significant. These results suggest that Selitrectinib can inhibit EN fusion-dependent interactions that leads to signaling pathway activation. These results show that inhibition is accompanied by lowered protein abundances, in case of the EN variants, making it more difficult to ascertain which effects are due to inhibition alone or the concomitantly smaller protein amounts.

### 3.6. A Framework for EN2 Activation of Key Signaling Pathways

Collecting the information from our various analyses, we sought to understand how the EN2 variant is linked to key signaling pathways in both normal and disease conditions (Figure 6). The EN2 variant interacts with components of several archetypal kinase signaling events: ERBB, insulin, JAK and BCR-ABL (chronic myeloid leukemia) based on KEGG enrichment analysis, which also share downstream signaling with ERK/ELK and PI3K/AKT signaling and also with JNK/AP-1 and STAT.

ERK signaling promotes proliferation and is often de-regulated in cancer. Our analysis show interactions with the common SHC1, GRB2, SOS complex, which link activated kinases to ERK signaling through RAS activation. In addition, IRS adaptors link insulin receptors to ERK signaling through SHC1 and SHIP2 phosphatase. Insulin receptor activity, in turn, is regulated by the EN2 interactor PTN1 phosphatase [84]. The BCR-ABL oncofusion also associates with ERK signaling through GRB2. The mid-ERK-signaling components were not detected as interactors, but phosphorylation of ERK and activity of the ERK-controlled transcription factor ELK were shown. ERK activity also phosphorylates translation initiation factors, through MNK, but the role in protein synthesis is not clear [85,86]. Notably, EN2 interacted with the eukaryotic initiation factor eIF-4E2, which is associated with translation under hypoxic conditions and is exploited in cancer [87,88]. EN2, like all the four insertless variants, also interacted with WDR83 (MORG1), which is an organizer of the ERK signaling pathway components [89].

PI3K phosphorylates phosphatidylinositols on the cytosolic side of the cytoplasmic membrane to recruit pleckstrin homology domain containing proteins of the cell membrane. A major phosphatidylinositol-recruited kinase is AKT (PI3K-AKT signaling), which promotes survivability and anti-apoptosis and in cancer is activated along with ERK signaling to provide a combined effect of proliferation and anti-apoptosis. EN2 interacted with phosphatidylinositol kinase PIKFYVE and also three phosphatidylinositol phosphatases: MTMR6, SYNJ1 and SHIP2. The SHC1-GRB2 interactions can also lead to PI3K/AKT activation by binding with GAB adaptors [67], which did not interact with EN2. The SHIP2 phosphatase can negatively regulate AKT signaling also by dephosphorylating GAB [90]. The insulin receptor regulating PTN1 phosphatase also inhibits AKT signaling [84]. Similarly, as with ERK pathway interactors, PI3K-AKT pathway mid-components were not considered as interactors, but proteins regulated by AKT phosphorylation were detected. GSK3 did interact with EN2, and GSK3 activity is suppressed by AKT [91]. EN2 also interacted with eIF-4E2 (eukaryotic initiation factor), which is released from the inhibition of eIF-4EBP by AKT, through MTOR [92]. AKT, through MTOR-activated S6K1, also induces phosphorylation of S6 (ribosomal protein S6), which was elevated in our stable cell line expressing EN2. AKT also activated MDM2 to degrade the tumor suppressor P53, which interacted with EN2. Both GSK3 and APC, involved in mediation of WNT signaling through beta catenin degradation, were also interactors of EN2.

Our data shows elevated STAT protein phosphorylation and transcriptional activity, which are canonically activated by JAK signaling. Only STAT1 interaction was observed with EN2, though, activation of STAT3 and STAT5 was more evident than that of STAT1. We found no interaction with EN2 to JAK or other components of the canonical JAK-STAT signaling pathway. Activation of STAT proteins might therefore happen through non-canonical STAT pathways. EN might also promote STAT activity by removing STAT inhibiting factors. Notably, the interactor SHIP2 is both negative and positive regulator of JAK signaling, and the interacting PTN1 phosphatase is a negative regulator of STAT signaling [93].

EN oncofusions have previously been associated with AP-1 [22] and increased expression of AP-1 dimer components have been observed in EN harboring MASC and leukemia [22,94]. We also found interactions with NCK1/2 adaptors associated EN2 with the JNK/AP1 pathway activation. It is notable that these interactors persisted even when EN2 was treated with Selitrectinib, which might suggest the interaction to not be completely dependent on EN2 phosphorylation. The activation of AP-1 was shown in luciferase assays with transient transfection, though NTRK3 and ETV6 increased AP-1 activity to similar levels. Selitrectinib treatment did show significant reduction in AP-1 activity in EN2 compared to NTRK3.

These pathways, which are clearly impacted by EN2, have been linked to many important cellular functions, such as cell proliferation, survival, protein synthesis and glucose regulation. The EN2 variant therefore, via interactions with many adaptor proteins that link kinases to their downstream components, has the potential to impact these important cellular functions. Activation of downstream components of these pathways provides support for the idea that these pathways are indeed being activated by EN2, with adaptor proteins likely acting as starting points for alteration of these pathways by EN2.

## 4. Discussion

Here, we present the largest to-date protein–protein interactome study of ETV6-NTRK3 oncofusions, including the signaling pathways they activate and the identification of new interactions of these pathways. Together, our data show that EN oncofusions have many interactors in common with several well-defined oncogenes.

Based on our extensive literature examination of a large amount of ETV6-NTRK3 related clinical articles, it is evident that the EN2 variant is generally the most common EN variant, while EN3 is dominant in thyroid carcinoma. The way EN is found in both solid and hematological malignancies was also reflected in their interactomes as enrichment of interactor associations also reveal terms with ERK and ERBB, which are common in solid malignancies, and leukemia-related terms. The SHC protein is an important mediator of several important signaling pathways, but its importance to EN-mediated oncogenesis is uncertain as EN2 is the most common variant, yet lacking the NTRK3 Y516 interaction site for SHC and stable interaction as well [30]. Though, in Ba/F3 cells, SHC phosphorylation is reportedly induced by EN4 expression, which also lacks the Y516 interaction site [37]. It is still uncertain whether EN, without the Y516, can phosphorylated SHC be directly recruited through IRS2 [32]. Our results show that the EN variants operate in close proximity with SHC, regardless if they have the known interaction site for SHC or not. Notably, the canonical variant EN2 did not activate ELK-1 (ERK) as strongly as EN1 and EN3, which retain the Y516 unlike EN2. This can cause differences between the variants as moderate activation of ERK can promote cell cycle progression, but strong activation of ERK can lead to cell cycle arrest [95]. All of the EN variants also strongly interacted with ARHG5 (also known as the breast cancer oncogene TIM), which activates JNK, RHOA and AP-1 through ROCK [96]. KIAA1671 interacted with all of the studied EN variants and had the highest interaction signals among all found interactors. KIAA1671, although being poorly characterized, has been associated with cancer in several occasions: antibodies against KIAA1671 are found in breast cancer patient sera [97], KIAA1671 deletion was also found to be among the most common structural variants in organoids derived from intraductal papillary mucinous neoplasms [98], KIAA1671 has emerged as a gene associated with carotid paragangliomas [99] and KIAA1671 has lowered expression in PTC (papillary thyroid carcinoma) [100], a malignancy where EN1 and EN3 are reported.

Knowing that these EN variants appear to have highly variable frequencies in cancer, we wanted to analyze the properties of these EN fusions and whether the EN2 variant distinguishes itself over others. The observed increase in pSTAT5 in the generated EN2 cell line was perhaps the most striking property that separated EN2 from the other variants. STAT5 is of relevance for EN2 in MASC, where STAT5 expression is reported [101,102] and shows elevated pSTAT5 [103], and also leukemia where STAT5 activation is driven by ABL oncogenes [104]. Our analysis also showed EN2 to also have the strongest association with ABL kinases, which could be a likely a culprit in STAT5 activation. However, there is also a study showing that the ABL inhibitor imatinib did not suppress the proliferation of an EN-expressing cell line [105]. Interacting CRK adaptors are also strongly implicated with ABL, though, these adaptors interacted with both EN+i and EN variants. The CRK adaptors are also linked to breast cancer anti-estrogen resistance proteins BCAR1 and BCAR3, which also associate with each other [106]. Both BCAR1/3 interacted with several EN+i variants but only EN2 of the insertless variants. BCAR3 is associated with inhibition of TGFbeta signaling [107], which is also previously reported for EN [108]. The APC tumor suppressor, frequently mutated in colorectal cancer [109], was also found to interact only with EN2. APC is a substrate of the GSK3A/B kinases, which interacted with EN2 and also with EN1 and EN3. EN2 interaction with both GSK3A/B and APC displays a stronger association with APC-mediated functions, than the other variants. APC has a role in the degradation of beta-catenin, which cooperates with WNT pathway transcription factor TCF in the canonical WNT pathway [110]. WNT pathway target genes have also been reported to be enriched in EN tumor samples [22]. One of APC functions is to halt the cell cycle by regulating expression of cyclin D1 and MYC, and conversely, over activated cyclin D1 and MYC can prevent APC from regulating the cell cycle [111]. Expression of MYC and cyclin D genes is commonly activated by STAT proteins [112], and pSTAT3 and pSTAT5 were higher for EN2 than the other variants. Evidence suggests that EN2 is still not unique in activating these pathways, for example, EN3 has been shown to cause IRS phosphorylation and AKT activation [55]. Lastly, SHIP2, while interacting with both EN2 and EN4 and all four of the EN+i variants, should not be overlooked, knowing the additional unique interactors possessed EN2. SHIP2 is a phosphatase with oncogenic properties, associated with activation of ERK, AKT, insulin and JAK-STAT signaling, but also has some tumor suppressor functions [113].

Based on our analysis, the canonical EN2 variant does appear to have unique interactors and properties, which can grant an edge in oncogenesis over the other variants. The factors indicated that elevated pSTAT3 and pSTAT5 and interactions with SHIP2, ABL and APC might contribute to the wider tissue portfolio of EN2. STAT5 activation alone can lead to activation of ERK and AKT pathways [114] and increased expression of Cyclin D1 [115], which is observed in both EN-expressing model cell lines [23,66] and EN tumors [22]. Though, even if EN2 has these additional properties, it does not rule out the possibility that it could be the most likely variant to occur through mechanisms that govern genomic translocations [116].

We found the NTRK3 kinase insert to notably affect the abundance of EN variants. For most of the variants, either the absence of the insert enabled higher protein abundance, or conversely, the insert caused a reduction in protein abundance. The EN protein abundance is likely regulated by ubiquitin-mediated degradation as EN degradation has previously been reported to occur through RNF123-mediated ubiquitinylation [117], though we found no interactions with RNF123. Unfortunately, the ubiquitin-mediated degradation of the NTRK family of kinases is not well studied, and only predictions exist for NTRK3, in which the likely mediators of NTRK3 degradation would be TRAF6 and SQSTM1 [118,119]. We observed no interaction with TRAF6 but SQSTM1 interacted with all the EN+i variants but none of the EN variants. However, EN2 also interacted with SQSTM1 upon inhibition. This is a potential support for SQSTM1’s involvement in EN degradation, as the inhibition also greatly reduced the abundance of EN2. SQSTM1 forms a regulatory trio with TRAF6, a K63 ubiquitin ligase and CYLD, a K63 deubiquitinase, which has been shown to control NTRK1 turnover [120]. CYLD interaction was observed with all of the studied constructs. The TAB3 adaptor, which interacted with all EN+i variants and EN2, also regulates the protein levels of SQSTM1 [121].

Overall, our study provides new insights into the mechanisms underlying the oncogenic activity of ETV6-NTRK3 oncofusions and has important implications for the development of new strategies for the treatment of cancers associated with these oncofusions.

## 5. Conclusions

Our study shows that EN oncofusions have rich interactomes with proteins relevant to oncogenesis, which can be detected by proximity labeling methods. Additionally, largely similar domain structures of oncofusion breakpoint variants can still yield significant differences in interactomes and their ability to activate cellular signaling pathways, which might explain for variant specific tissue and cancer associations. Inhibition of these variants leads to a reduction of their protein abundance, which suggests their targeting for proteasomal degradation.

## Figures and Tables

**Figure 1 cancers-15-04246-f001:**
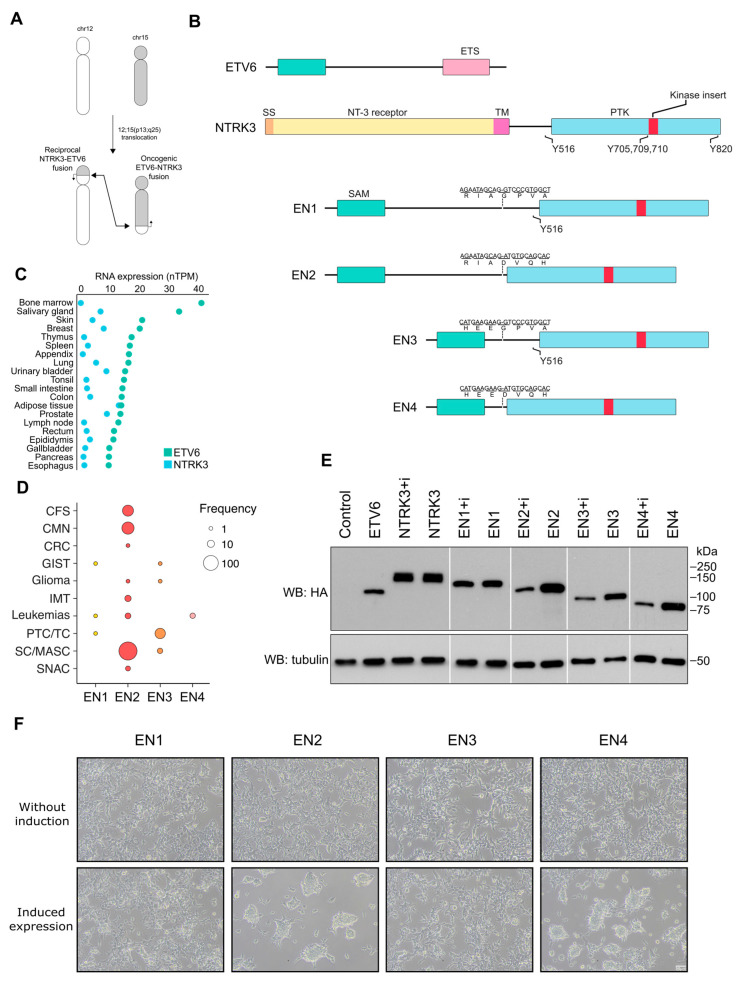
ETV6-NTRK3 (EN) fusion variants, their frequencies in cancer and their effects on generated HEK293 cell line morphology. (**A**) Schematic of chromosomal rearrangements resulting in the formation of the ETV6-NTRK3 fusion gene. (**B**) ETV6 and NTRK3 expression levels in various tissues, as shown by the protein atlas database. (**C**) Domain structures of wild-type ETV6, NTRK3 and EN break point and kinase insert splice variants: EN1 (EN e5–e12), EN2 (EN e5–e13), EN3 (EN e4–e12) and EN4 (EN e4–e13). Break point sequences of EN variants are shown above the fusion structures. (**D**) Frequencies of EN variants reported in cancer samples, grouped by cancer type: CFS (congenital fibrosarcoma), CMN (congenital mesoblastic nephroma), CRC (colorectal carcinoma), GIST (gastrointestinal stromal tumor), Glioma (pediatric low-grade glioma, pediatric high-grade glioma), IMT (inflammatory myofibroblastic tumor), Leukemia (acute myeloid leukemia, acute promyeloid leukemia, acute lymphoblastic leukemia, Ph-like acute lymphoblastic leukemia, chronic eosinophilic leukemia), PTC (papillary thyroid carcinoma/thyroid cancer), SC (secretory carcinoma) and SNAC (sinonasal adenocarcinoma). Variants are color coded for clarity. (**E**) Expression levels of the studied EN variants in stable, inducible cell lines, as determined using lysate analysis. The control sample is the EN2 cell line lysate without expression induction. (**F**) Morphology of the stable cell lines with and without tetracycline-induced EN variant expression. Scale bar: 50 μm. See Appendix A for original Western Blots.

**Figure 2 cancers-15-04246-f002:**
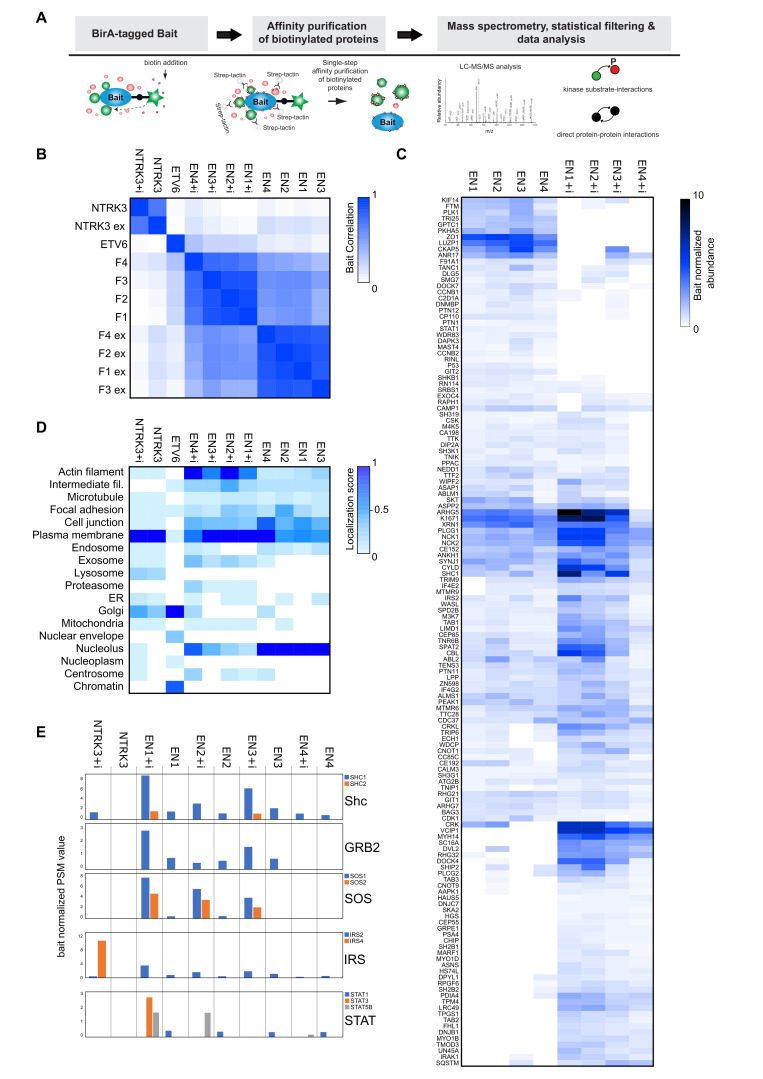
The interactome of NTRK3, ETV6 and EN variants. (**A**) Schematic view of the pipeline used to analyze protein–protein interactions. (**B**) Correlation plot clustering of the studied constructs by their interactomes, showing the relationships between the different bait proteins. (**C**) Estimates of the subcellular localization of the studied constructs, based on their protein interactomes and the interactomes of corresponding subcellular markers. (**D**) Comparison of previously reported and studied interactors of EN variants. (**E**) Comparison of the interactomes of EN and EN+i variants, showing only those interactors that interacted with all four variants in at least one group.

**Figure 3 cancers-15-04246-f003:**
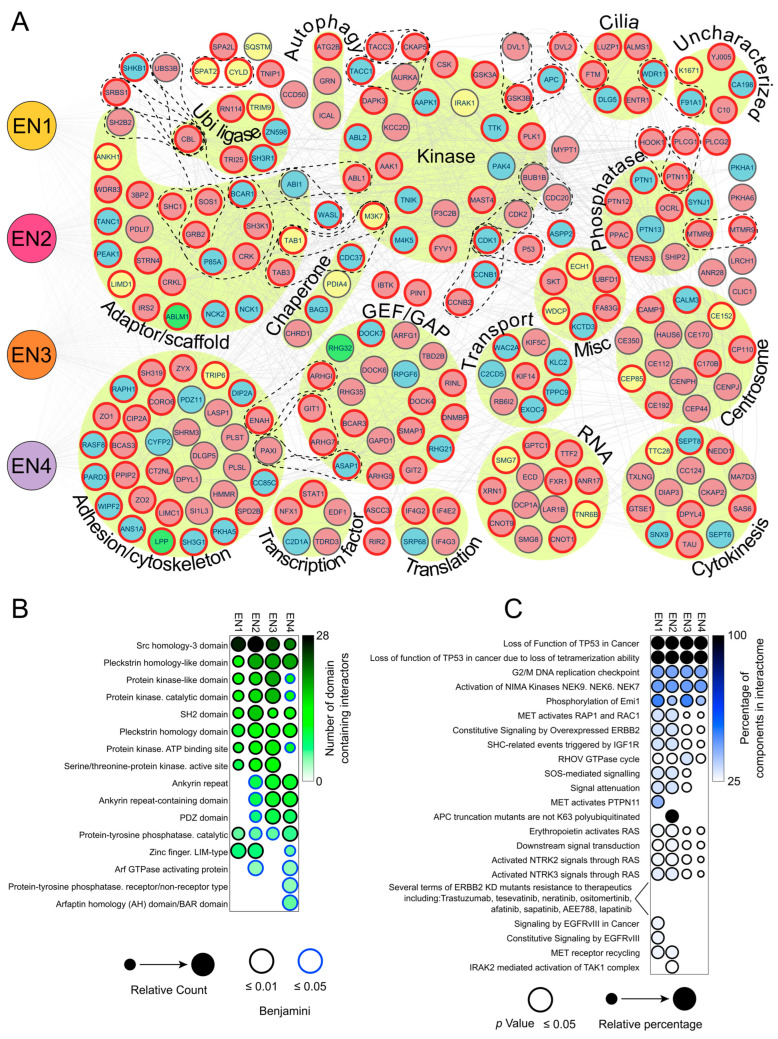
Interactome map and comparison among EN variant interactomes. (**A**) Nodemap showing all of the four EN variant interactors, grouped by their gene ontology (biological processes) and the complexes they form (according to the CORUM database). The nodes are colored according to their protein interactions: blue for interaction with both NTRK3 and EN fusions, yellow for interaction with ETV6 and EN fusions, green for interaction with NTRK3, ETV6 and EN fusions and red for interaction with EN fusions only. The interactors of the canonical EN2 variant are highlighted with red borders. (**B**) Dotplot comparison of the protein domains in the interactors, as determined by the DAVID InterPro database (benjamini *p* < 0.05). (**C**) Dotplot of Reactome terms enriched from all of the EN variant interactomes. Only terms with over 25% of known components found in the interactome are shown (*p* < 0.05).

**Figure 4 cancers-15-04246-f004:**
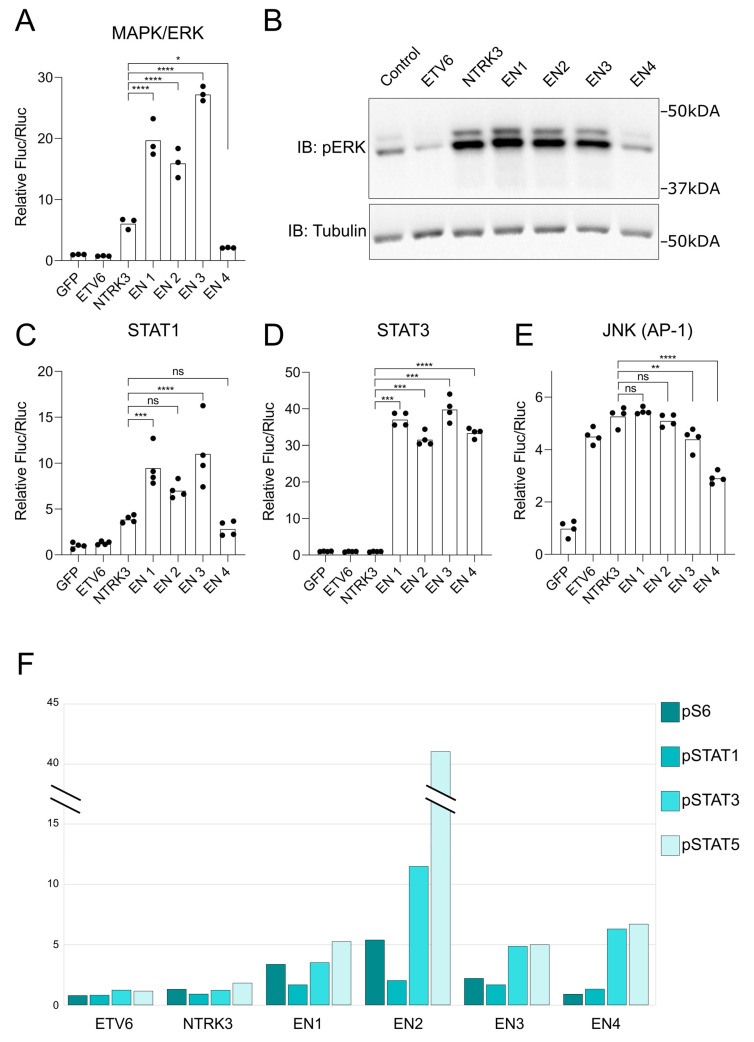
Activation of signaling pathways by EN variants and their potential mechanisms in oncogenic transformation. (**A**) Activation of the ERK pathway, assessed using a transcription factor ELK1-responsive luciferase reporter. (**B**) Western blot quantification of ERK1/2 (T202/Y204) phosphorylation levels, 24 h post-induction in generated stable cell line lysates, revealing the intensity of ERK phosphorylation across different EN variants. (**C**) Activation of the STAT1 signaling pathway determined using a luciferase assay with a STAT1-responsive element. (**D**) Activation of the STAT3 signaling pathway assessed using a luciferase reporter assay with a STAT3-responsive element, indicating the potential role of EN variants in STAT3-mediated cellular processes. (**E**) Assessment of the JNK pathway activation by EN variants, using an AP-1-responsive luciferase reporter assay. (**F**) Mass cytometry-derived mean values of phosphorylation activation sites of ribosomal protein S6 (S235/S236), STAT1 (Y701), STAT3 (Y705) and STAT5 (Y694) detected at the single-cell level in stable cell lines, normalized to a control. The data show the relative intensity of signaling events initiated by each EN variant. The EN2 cell line without tetracycline induction was employed as the reference control for the normalization of stable cell line experiments. Luciferase assay column graphs individual measurements are shown as black dots and the level of statistical significance of the EN variant values compared to NTRK3 values, is denoted by asterisk-symbols: * (*p* < 0.05), ** (*p* < 0.005), *** (*p* < 0.0005), **** (*p* < 0.00005) or “ns” for not significant (*p* > 0.05). See Appendix A for original Western Blots.

**Figure 5 cancers-15-04246-f005:**
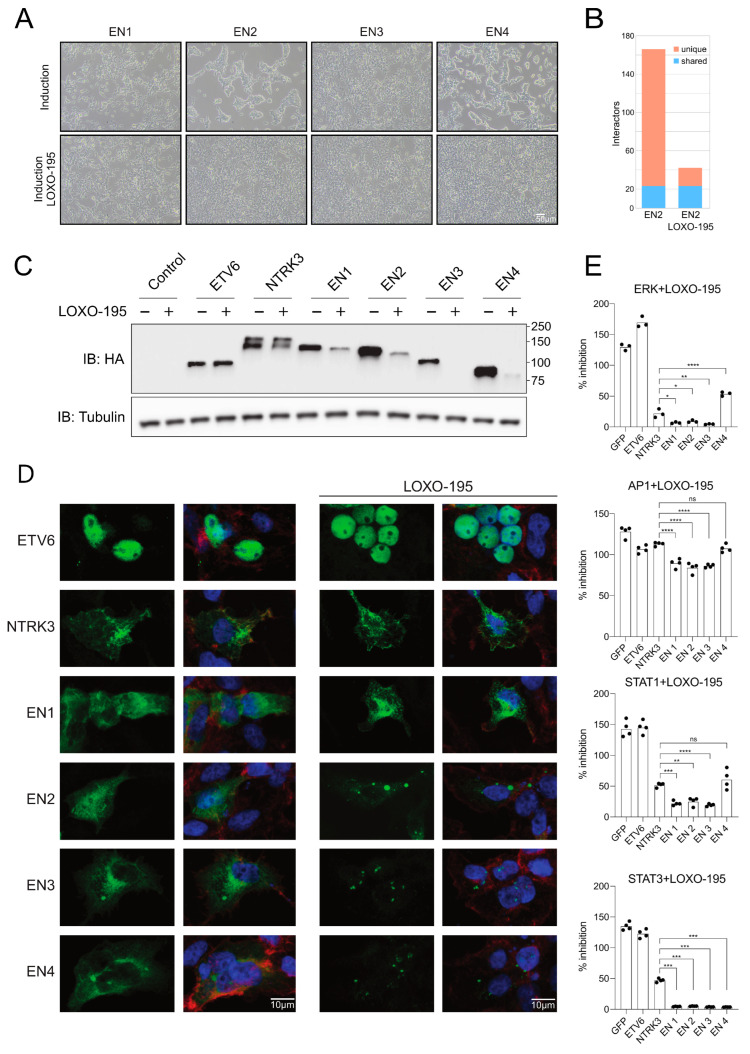
Detailed examination of Selitrectinib (LOXO-195) influence on EN variant behavior, localization and downstream signaling. (**A**) Morphological analysis of stable cell lines expressing different EN variants in the absence and presence of 100 nM Selitrectinib. The visual differences highlight the inhibitor’s impact on stable cell lines morphology by EN variants. Scale bar: 50 μm. (**B**) A bar graph comparing the BioID interactors of the EN2 variant in the absence and presence of Selitrectinib. The interactors that were shared by both conditions are represented in blue, while the unique interactors for each condition are showcased in orange, indicating the molecular partners potentially affected by Selitrectinib. (**C**) Western blot analysis showing the influence of Selitrectinib on the protein levels of EN variants in stable cell lines. (**D**) Fluorescence microscopy images portraying the subcellular localization of EN variants under two conditions: cells untreated and cells treated with Selitrectinib. The green color shows the location of the studied proteins on the left and the picture on the right show the location together with the blue DAPI stained nucleus and red stained actin cytoskeleton. Scale bar: 10 μm. (**E**) Luciferase assay results detailing the impact of Selitrectinib on the activation of four key signaling pathways, namely, ERK (via ELK1 reporter), STAT1, STAT3 and JNK (via AP1 reporter). The readouts are indicative of the inhibitor’s effect on these crucial oncogenic pathways driven by EN variants, with reporter expression levels presented relative to untreated cells. Luciferase assay column graphs individual measurements are shown as black dots and the level of statistical significance of the EN variant values compared to NTRK3 values, is denoted by asterisk-symbols: * (*p* < 0.05), ** (*p* < 0.005), *** (*p* < 0.0005), **** (*p* < 0.00005) or “ns” for not significant (*p* > 0.05). See Appendix A for original Western Blots.

**Figure 6 cancers-15-04246-f006:**
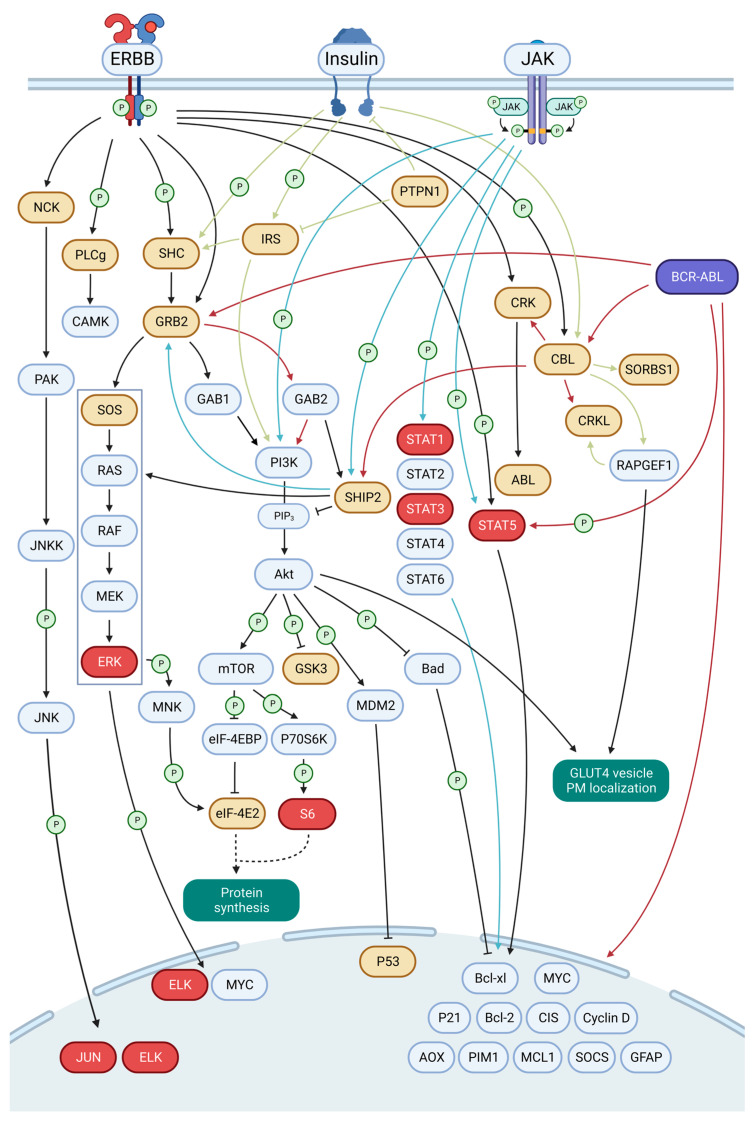
Association of EN2 with ERBB, insulin, BCR-ABL and JAK signaling. This figure shows EN2 interacting pathway components in orange and proteins that were shown to be activated in red. The other the pathway components are shown in gray. The signaling pathways were adapted from the KEGG pathways database. The first arrows starting from the different signaling kinases are color coded and sharp arrow heads indicate activating interactions and blunt arrow heads indicate inhibiting interactions and the arrows with the letter “P” indicate interactions which involve phosphorylation events.

## Data Availability

The datasets produced in this study are available in the following database: MassIVE (https://massive.ucsd.edu/) with web access MSV000091061. Source data have been provided as Source Data files. All other data supporting the findings of this study are available from the corresponding author on reasonable request.

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
