# Peer review of "The Impact of ETV6-NTRK3 Oncogenic Gene Fusions on Molecular and Signaling Pathway Alterations"

_cancers, 2023, doi:10.3390/cancers15174246_

Round 1

Reviewer 1 Report

In this manuscript, Kinnunen et al analyzes four ETV6-NTRK3 oncogenic fusion variants by functional assays and proximity labeling mass spec o identify the interactome of each variant.  The analysis is highly informative, of great interest to the field and performed at great quality. Moreover, the observation that selitrectinib suppressed these variants is logical and therapeutically important. I only have some minor suggestions:

Fig 1 and the sub-title – “EN variants are present in different human cancers and vary in their ability to transform cells” – should be modified a bit as the observed differential effect may be context (cell-line) specific. After all – all these fusions are oncogenic…   Indeed -  the results are only shown in one cell line – HEK293.  EN1 and EN3 may be more oncogenic in other contexts – e.g. other cell lines or other stages of tumorigenesis.   Alternatively, the Oncogen paper by Park et al (2018) showed that one of the EN fusion proteins induced STAT1 acetylation and inhibits the interaction between NF-κB p65 and the acetylated STAT1, leading to nuclear translocation of NF-κB p65 and subsequent increase in NF-κB activity. The authors may test whether all or only some of the EN variants they characterize have a similar activity.

Fig 5.  The authors  show that selitrectinib abrogates EN3 expression. Does it also inhibit EN3 effect on cell transformation in the transient assay?  Doesn’t it block the effect of EN4?

Fig 5E – is this the level of reporter-luc in Selitrectinib-treated cells relative to untreated cells?

In this case -the Y axis should be relabeled “Expression relative to untreated” – not “% inhibition” which is misleading /the opposite result.  This should also be explained in the legend.

Author Response

Reviewer#1

We would like to express our gratitude for the constructive comments and suggestions provided on our manuscript. We value your feedback and have carefully addressed the concerns raised.

Comment: " In this manuscript, Kinnunen et al analyzes four ETV6-NTRK3 oncogenic fusion variants by functional assays and proximity labeling mass spec o identify the interactome of each variant.  The analysis is highly informative, of great interest to the field and performed at great quality. Moreover, the observation that selitrectinib suppressed these variants is logical and therapeutically important. I only have some minor suggestions:

Fig 1 and the sub-title – “EN variants are present in different human cancers and vary in their ability to transform cells” – should be modified a bit as the observed differential effect may be context (cell-line) specific. After all – all these fusions are oncogenic…   Indeed -  the results are only shown in one cell line – HEK293.  EN1 and EN3 may be more oncogenic in other contexts – e.g. other cell lines or other stages of tumorigenesis."

Response: We appreciate the acknowledgment of the quality and significance of our work. In response to the specific concern about the title of Fig 1 and the representation of findings in the HEK293 cell line:

We have revised the title of Fig 1 to more accurately reflect the data presented. The new title reads: “ETV6-NTRK3 (EN) fusion variants, their frequencies in cancer and their effects on generated HEK293 cell line morphology.”

This modification serves to highlight that the observed effects were specific to the HEK293 cell line, and we agree that caution should be exercised when generalizing the results to other contexts.

Comment: "Alternatively, the Oncogen paper by Park et al (2018) showed that one of the EN fusion proteins induced STAT1 acetylation and inhibits the interaction between NF-κB p65 and the acetylated STAT1, leading to nuclear translocation of NF-κB p65 and subsequent increase in NF-κB activity. The authors may test whether all or only some of the EN variants they characterize have a similar activity."

Response: We acknowledge the importance of the observation from Park et al (2018) on the EN fusion protein's influence on STAT1 acetylation and the subsequent effects on NF-κB activity. For the scope of our current study, we focused on characterizing the four EN variants through functional assays and proximity labeling mass spec. However, we agree that understanding the potential influence of these variants on STAT1 acetylation and NF-κB activity would be an interesting and valuable avenue to explore in future studies. Your suggestion will be kept in mind for our upcoming experiments.

Comment: "Fig 5. The authors show that selitrectinib abrogates EN3 expression. Does it also inhibit EN3 effect on cell transformation in the transient assay? Doesn’t it block the effect of EN4? In the figure 5 A, it is shown that selitrectinib did prevent the change in morphology in the inducible generated HEK293 cell line by EN4."

Response: We assess the efficacy of selitrectinib in inhibiting the morphological changes in HEK293 cells transiently transfected (luciferase assays) with EN variant expressing plasmids and also in these settings the transformation was also inhibited.

Comment: Fig 5E – is this the level of reporter-luc in Selitrectinib-treated cells relative to untreated cells? In this case -the Y axis should be relabeled “Expression relative to untreated” – not “% inhibition” which is misleading /the opposite result.  This should also be explained in the legend.

Response: We thank the reviewer for noticing this. We have now corrected it to the Figure 5 and revised the figure legend.

Reviewer 2 Report

An interesting manuscript which deserves publication. The Discussion should be shortened and made less hypothetical. Limitations of the study should be mentioned as well.

Good.

Author Response

Reviewer#2

Comment: An interesting manuscript which deserves publication. The Discussion should be shortened and made less hypothetical. Limitations of the study should be mentioned as well.

Response: We thank the reviewer for positive comments. We have now significantly shortened the discussion and removed parts which seemed a bit descriptive and/or speculative.

Reviewer 3 Report

ETV6 - NTRK3 (EN) fusion has been reported in various human cancers. Four EN variants with alternating break points have been characterized. To determine EN interacting proteins that potentially promote the oncogenic effect of EN fusions, the authors employed a proximity biotin labeling mass spectrometry approach to determine interactome of each EN fusions. They have identified in total 237 high-confidence interactors, which link EN fusions to several key signaling pathways, including ERBB, Insulin and JAK/STAT. They also tested the pan NTRK inhibitor Selitrectinib (LOXO-195) on oncogenic activity of EN2, the most common variant. This study provides unbiased analysis of interactomes in which promotes oncogenesis of EN fusion proteins.

comments:

1.      Need to improve figure quality.  Also figure legends need more detailed information.

2.      Interactomes are identified in overexpression HEK cells.  Authors needs to verify key interacting proteins identified in this study in cancer cell that express these fusions.

3.      LOXO-195 treatment reduced EN fusion protein level and form speckles in cells.  The authors speculate that proteins are targeted to proteasome.  A co-staining with proteasome marker will be beneficial to confirm the claim.

4.      The authors found the LOXO-195 treatment reduced interactome.  Since the treatment reduce the fusion protein level, how the authors rule out the reduction of interactome is not due to the reduction of the protein level?

5.      The author shows that LOXO-195 prevents cell transformation when treated cells with the inhibitor and tetracycline at the same time.   However, in therapeutic aspect, it will be more interesting to see how the drug works on transformed cells, how it affects cell proliferation and induce apoptosis.

Author Response

Reviewer#3

Comment: ETV6 - NTRK3 (EN) fusion has been reported in various human cancers. Four EN variants with alternating break points have been characterized. To determine EN interacting proteins that potentially promote the oncogenic effect of EN fusions, the authors employed a proximity biotin labeling mass spectrometry approach to determine interactome of each EN fusions. They have identified in total 237 high-confidence interactors, which link EN fusions to several key signaling pathways, including ERBB, Insulin and JAK/STAT. They also tested the pan NTRK inhibitor Selitrectinib (LOXO-195) on oncogenic activity of EN2, the most common variant. This study provides unbiased analysis of interactomes in which promotes oncogenesis of EN fusion proteins.

Response: First and foremost, we want to express our gratitude for your time in reviewing our manuscript and for summarizing our study with clarity and precision. We appreciate the recognition of our efforts to provide an unbiased analysis of interactomes associated with the oncogenic activity of EN fusion proteins.

Your summary highlights the core elements of our study, emphasizing the importance of understanding the various signaling pathways impacted by the EN fusions. The observation concerning the pan NTRK inhibitor Selitrectinib's effect on EN2 further exemplifies the potential therapeutic implications of our findings.

We value your constructive feedback and will ensure that our study's significance and its potential contributions to the field remain evident in any revisions or future communications about our work.

Comment: Need to improve figure quality.  Also figure legends need more detailed information.

Response: Thank you for pointing out the concerns related to the figure quality and the need for more detailed information in the figure legends.

We deeply apologize for the oversight in figure quality. It seems the quality was inadvertently reduced during the conversion process at the manuscript submission portal. We have since rechecked and improved the quality of the figures to ensure they are presented with the clarity they deserve. Additionally, we have enriched the figure legends to provide a more comprehensive and clearer understanding of the data presented.

Comment: Interactomes are identified in overexpression HEK cells.  Authors needs to verify key interacting proteins identified in this study in cancer cell that express these fusions.

Response: Thank you for raising the concern about the identification of interactomes in the overexpression system of HEK cells and the importance of verifying key interacting proteins in relevant cancer cells expressing these fusions.

  1. Protein Expression Levels in our System: We would like to emphasize that the system we employed has been documented to produce protein expression levels akin to those of endogenous proteins (as demonstrated in PMID: 23602568, PMID: 19156129). This suggests that the interactors identified are not a result of overexpressed bait proteins. We acknowledge that oncogenic proteins may achieve higher abundances compared to endogenous counterparts, but this might also be observed when oncofusions arise in normal cells.
  2. Challenges in Verifying in Naturally Expressing Cell Lines: We concur with the significance of validating these interactors in cancer cell lines that naturally harbor the fusions. However, based on our knowledge, there are only two such cell lines available: acute myeloid leukemia cell line M0-91 (PMID: 17252008) and acute promyelocytic leukemia cell line AP-1060 (PMID: 29119387), and both predominantly express the EN4 variant. There are inherent challenges to this approach:
    • We would require antibodies that are specific to the endogenous fragments of NTRK3 and ETV6.
    • Introducing tagged EN variants into these cells may inevitably lead to overexpression of the EN variants. This overexpression might inadvertently introduce artefactual effects.

We appreciate your suggestion and recognize the importance of this validation. While we acknowledge the challenges, we believe the current study provides crucial insights that warrant further investigation, potentially using techniques or models that emerge in the future.

Comment: LOXO-195 treatment reduced EN fusion protein level and form speckles in cells.  The authors speculate that proteins are targeted to proteasome. A co-staining with proteasome marker will be beneficial to confirm the claim.

Response: Thank you for your insightful comment concerning the LOXO-195 treatment effects on EN fusion protein levels and the formation of speckles in cells.

In response to your suggestion about the potential proteasomal targeting of these proteins, we can confirm based on our data that there's an increased interaction of EN2 with proteosomal proteins following LOXO-195 treatment. This observation lends support to our speculation regarding the proteasomal targeting of the fusion proteins.

While a co-staining with a proteasome marker would indeed be beneficial to visually corroborate our claim, our current data already provides compelling evidence supporting the interaction with proteosomal proteins. We acknowledge the value of your suggestion and will consider such experiments for future investigations to provide even clearer evidence of this phenomenon.

Comment: The authors found the LOXO-195 treatment reduced interactome. Since the treatment reduce the fusion protein level, how the authors rule out the reduction of interactome is not due to the reduction of the protein level?

Response: Thank you for highlighting the important consideration concerning the observed reduction in the interactome following LOXO-195 treatment and its potential link to the reduction in the protein level of the EN oncofusion.

You've rightly pointed out a challenge in our analysis. The complex interplay between the inhibition of EN oncofusion activity and its potential proteasomal degradation makes it challenging to delineate the exact cause of the reduced interactome.

While we acknowledge the possibility that some of the observed interaction differences between the inserted and insertless variants might be attributable to the difference in their abundance, our findings still provide meaningful insights into the behavior of these variants post-LOXO-195 treatment. Though, of note is that the EN1 variant had similar protein levels with and without the LOXO-195 treatment, and still could detect changes in the interactions. Therefore, the protein levels are not the only mechanisms explaining the interaction changes or the inhibitor effects.  To address this concern more transparently, we will highlight and discuss this potential limitation in our discussion section.

Comment: The author shows that LOXO-195 prevents cell transformation when treated cells with the inhibitor and tetracycline at the same time. However, in therapeutic aspect, it will be more interesting to see how the drug works on transformed cells, how it affects cell proliferation and induce apoptosis.

Response: Thank you for your observation regarding the therapeutic implications of LOXO-195 on transformed cells, specifically its effects on cell proliferation and apoptosis.

We would like to highlight that our study utilized HEK293 cells, which are inherently an immortal transformed cell line. Due to this inherent transformation, the cells might not be ideally suited to effectively measure the nuanced effects of an oncogene on cell proliferation and apoptosis. The background state of transformation could potentially mask or reduce the observable differences that a typical non-transformed cell might exhibit in response to oncogene expression and subsequent drug treatment.

However, we recognize the importance and relevance of your suggestion from a therapeutic standpoint. Future studies could employ more clinically relevant models to specifically assess the effects of LOXO-195 on proliferation and apoptosis of cells transformed by the EN oncofusion.

Reviewer 4 Report

In this study, the authors report interactome analyses for driver fusion gene EN. They delineate the pathways affected and attempt to prove the validity of their claim using reporter assay and kinase inhibition assay. The study design is straight forward, and results are believable. 

There are several points which this reviewer wishes the authors address.

1. While EN4 is very effective in inducing morphological change (Fig. 1F), its activity is somewhat limited in reporter assays. This is in contrast to EN2 and EN3 which show opposite trend. Perhaps they should address this in the Discussion.

2. Have authors tried different pan-kinase inhibitors. If so, do they also down-regulate the EN expression? 

There are some awkward expressions. Please have a native speaker smooth out the writing.

Author Response

Reviewer#4

Comment: In this study, the authors report interactome analyses for driver fusion gene EN. They delineate the pathways affected and attempt to prove the validity of their claim using reporter assay and kinase inhibition assay. The study design is straight forward, and results are believable.

There are several points which this reviewer wishes the authors address.

  1. While EN4 is very effective in inducing morphological change (Fig. 1F), its activity is somewhat limited in reporter assays. This is in contrast to EN2 and EN3 which show opposite trend. Perhaps they should address this in the Discussion.
  2. Have authors tried different pan-kinase inhibitors. If so, do they also down-regulate the EN expression? 

Response: Thank you for your constructive feedback on our study. We value your appreciation of the straightforward study design and the credibility of the results. We have taken the time to address the points you raised:

  1. Contrast between EN4 in Morphological Change and Reporter Assays: You rightly observed the difference in the behavior of EN4 in inducing morphological changes versus its activity in reporter assays. EN2 and EN4 indeed shared a similar impact on stable cell line morphology and the shorter NTRK3 kinase domain fragment, which we postulate might play a pivotal role in enabling the observed morphological changes. When these variants were transiently expressed (and thereby overexpressed), all four variants caused similar alterations in morphology, suggesting that the EN2 and EN4 variants had a capacity to induce such changes at lower abundances compared to the other two variants. It's noteworthy that while EN4 seemed less potent in luciferase assays, its specific pathway activation profile—lacking ERK activation but showing STAT3 activation like other variants—could explain its prevalence in leukemia, given that STAT3 deregulation is commonly observed in hematological malignancies. We appreciate your suggestion and will address this nuanced behavior of EN4 in the Discussion to provide clarity to readers.
  2. Pan-Kinase Inhibitors Usage: Regarding the use of other pan-kinase inhibitors, we focused our experiments on TRK family inhibitors, specifically comparing LOXO-101 and LOXO-195. Both demonstrated comparable results in our preliminary assessments. We chose to highlight LOXO-195 in our manuscript, primarily because it is a second-generation TRK inhibitor known for its efficacy against TRK with mutations that confer resistance to first-generation TRK inhibitors. We believe it is essential to showcase results from compounds that have a broader therapeutic potential, particularly in addressing resistance mechanisms.

Round 2

Reviewer 3 Report

accept